# COVID-19 Stress and Teachers Well-Being: The Mediating Role of Sense of Coherence and Resilience

Girum Tareke Zewude [1,2,*], Sisay Demissew Beyene [3], Belayneh Taye [4], Fatiha Sadouki [5] and Maria Hercz [6]

1    Doctoral School of Education, University of Szeged, 6724 Szeged, Hungary
2    Department of Psychology, Wollo University, Dessie 1145, Ethiopia
3    Department of Economics, Arsi University, Assela 193, Ethiopia
4    Department of Political Science and International Studies, Bahirdar University, Bahir Dar 79, Ethiopia
5    Doctoral School of Linguistics, University of Szeged, 6720 Szeged, Hungary
6    Faculty of Primary and Pre-School Education, Eötvös Loránd University, 1126 Budapest, Hungary
*    Correspondence: zewude.girum.tareke@edu.u-szeged.hu

**Abstract:** The outbreak of the COVID-19 pandemic has impacted many professions with short-, medium-, and long-term consequences. Hence, this study examined the mediating role of sense of coherence (SOC) and resilience in the relation to COVID-19 stress and teachers' well-being (TWB). It recruited 836 teachers from Ethiopia's higher-education institutions, of which 630 (75.4%) were men and 206 (24.6%) were women, with a mean age of 32.81 years and a standard deviation of 6.42. Findings showed that COVID-19 stress negatively predicted SOC, resilience, and TWB and that SOC and resilience positively predicted TWB. It was concluded that SOC and resilience, both together and separately, mediated the relation between COVID-19 stress and TWB. These results were discussed alongside relevant literature, and the study is found to be valuable for practitioners and researchers who seek to improve well-being using SOC and resilience as resources across teaching professions.

**Keywords:** COVID-19 stress; higher education teachers; mediation analysis; resilience; sense of coherence; teachers' well-being





## 1. Introduction

Since its outbreak in Wuhan, China, in late December 2019, the COVID-19 disease has affected virtually all people worldwide. As of 19 September 2022, there have been around 618.2 million positive cases, 6.5 million deaths, and 598.1 million recovered patients [1]. As a result, different countries were forced to implement World Health Organization (WHO) emergency protocols, which include limitations on nonessential individual movements and social activities. Meanwhile, thousands of critically ill COVID-19 patients are currently in hospital, and many families have lost their relatives [2].

The pandemic has also affected physical and mental health and societal well-being [3–5], triggered a socioeconomic crisis, and inflicted profound psychological distress on people worldwide [2]. It has also altered societal living conditions, which became a challenge to health experts' agenda [6] affected well-being [2], socioeconomic conditions, and the education system [7,8], and increased cases of suicide [9].

In the outbreak of infectious disease, frontline health workers [3] and teachers at all school levels worldwide are significantly affected [10–12]. The pandemic's impact has been substantial, especially on education. For instance, a study on how the COVID-19 pandemic has affected tertiary education students in Bangladesh revealed many unexpected interruptions in students' learning; low motivation; and economic, physical, and mental problems [11]. Specifically, the effect of this crisis on higher education has been an overlooked but potentially important issue [13], with profound outcomes among frontline workers of higher-education institutions [1]. The closure of universities worldwide and

the implementation of learning, teaching, and assessment on online platforms have caused changes to teachers' well-being [1].

The COVID-19 disease has had an adverse effect on teachers' well-being (TWB) globally, a profound issue that is expected to lead to short-, medium-, and long-term consequences for different actors and organizations [13]. Stress among teachers can be caused by amplified media exposure, the implementation of school closures, social distancing, and home quarantine [14]; and the stoppage of face-to-face teaching in higher education [13]. According to the United Nations Educational Scientific and Cultural Organization (UNESCO) [15], hundreds of millions of students, teachers, and national education planners have felt the impact of COVID-19, which has not been immediately visible but is expected to surface in the medium and long term [13]. A study conducted in the Philippines showed that more than half of Filipino teachers suffered from moderate COVID-19 stress, found that health status had a negative relation with COVID-19 stress, and observed that the participants experienced greater stress associated with the COVID-19 pandemic [12].

Ethiopia, the second most populous country in Africa, has also been greatly affected by the COVID-19 pandemic [16]. The first COVID-19-positive case in the country was officially confirmed on 13 March 2020, and in April 2020, schools, which were considered the most vulnerable sector, officially closed [17]. The effects of the outbreak were felt throughout the education sector, which discontinued teaching/learning activities for more than six months after the peak of the outbreak's first phase. In October 2020, the government reopened schools and implemented preventive measures as recommended by the WHO. Higher-education students, especially those at the PhD and master levels, attended classes by following social distancing protocols, wearing masks, and using sanitizers, and sometimes continued their education through online platforms such as Zoom, e-mail, and Skype. For teachers in universities with poor infrastructure access, such as in Africa, these new technologies can be a problem. Only a few universities in Ethiopia [11,13] had worked with online platforms such as Skype, e-mail, and Zoom [11,18].

In Ethiopia, only a few studies have focused on the COVID-19 pandemic. These studies have examined perceived work-related stress and associated factors among public secondary school teachers [19], the validity of the Fear of COVID-19 Scale in the Amharic language [20], the impact of COVID-19 on private higher education [16], and perceived stress and its associated factors among healthcare workers [21]. Unlike these studies, our research is unique in that it investigates the integrated novel framework of the positive emotion, engagement in life and work, positive relationships, meaning in life, and work accomplishments (PERMA) positive well-being model [22], the broaden-and-build theory of positive emotions [23], the resilience theory [24]; and the impact of COVID-19 on higher education [13]. In today's higher-education sector, examining how TWB is influenced by COVID-19 stress and protected by one's sense of coherence (SOC) and resilience is relevant. These studies have established a strong scientific groundwork, but they have yet to conduct a further psychometric inquiry on the Perceived Stress Scale of COVID-19, the Sense of Coherence scale, the Brief Resilience Scale, the PERMA-Profiler Questionnaire, and the mediation role of resilience and SOC between COVID-19 and TWB in higher education is crucial. Hence, this study aimed to determine the possible positive psychological resources (resilience, SOC, the PERMA positive well-being model) to manage COVID-19 stress and work during the pandemic. The following section explains the scientific evidence of the current study's conceptual model and their respective relations.

### 1.1. The Relation between COVID-19 Stress and Well-Being

Recent literature has shown a decline in TWB throughout the COVID-19 pandemic period [25,26]. The pandemic has had a significant impact on well-being, causing people to experience anxiety, fear, and stress [27]. In line with this, ref. [25] added that all teachers have been worried about their families' health and well-being during the pandemic. Additionally, their longitudinal research using a French sample [28] found that the

COVID-19 pandemic influenced people's well-being and relationships in many ways and was negatively associated with well-being [29].

Nowadays, as millions of people worldwide are beginning to overcome the isolation caused by the pandemic, the development of positivism plays many crucial roles in one's mental health [30]. Indeed, ref. [31] argued for a possible positive association between the COVID-19 pandemic and the PERMA model. It has been established that negative and positive emotions are two sides of the same coin and are everyday events. Some studies have assessed the proportion of negative and positive emotions experienced by an individual [32]. People who exhibit more positive emotions than negative emotions flourish in life and are satisfied with it, feel a sense of fulfillment, and could effectively recover from stressful situations [22,32]. In this regard, positive psychology, pioneered by Martin Seligman the founder of positive psychology is the scientific study of leading a meaningful life [33]; to increase individual happiness [33,34] and lower employee stress [35]. Currently, Seligman's modern positive psychology theory has a considerable impact on health workers [36,37], teachers [38,39], organizations [40,41], and individuals [42,43].

Based on the above evidence, this study proposed the PERMA positive well-being model, a multidirectional construct, as a framework to evaluate the nature of teachers' work-life balance to lower their stress levels during the pandemic. A study by [22] argued that human success or pleasure is ultimately the result of the interactions and capabilities of the five pillars of the PERMA model (positive emotion, engagement in life and work, positive relationships, meaning in life, and work accomplishments) [44].

### 1.2. The Relation between COVID-19 Stress, Resilience and SOC

The COVID-19 pandemic has had an adverse effect on people's psychological well-being, which is complicated by the fact that teaching is one of the most stressful professions [45,46]. The authors of [45] found that resilience and SOC are the best personal resources that may safeguard ego strength and lower stress levels and are essential during the COVID-19 pandemic. They are also the most critical psychological constructs that substantially support an individual's well-being and functioning level while under severe stress [45–48].

Resilience refers to an individual's mental strength and involves their ability to adapt to or overcome adversity or stress [45,49]. Several studies have explored the benefits of resilience, such as reduced individual COVID-19-related stress [47], a negative association with the fear of COVID-19, a positive impact on life satisfaction [48], physical and psychological adjustment [45,49], enhanced positive emotions rather than negative ones [50], increased happiness, lower stress levels, better recovery from symptoms of schizophrenia and depression [45], and disease resistance [51]. Remarkably, individuals who display altruistic behavior, positive emotions, hope, and cognitive flexibility have been described under challenging conditions throughout history and have been associated with resilience methods for overcoming adversity [47,52].

SOC is another considerable positive personal resource for teachers to overcome stress during the pandemic. According to [53] SOC pertains to an individual's toughness and capacity to respond to adverse situations. It also refers to a stable disposition across one's life span [54] that could help professionals understand the situation as clear and reasonable, adaptable, and meaningful, which enables their resilience [45]. Scholars have observed a negative relation between COVID-19-related traumatic distress and SOC [55,56]. SOC can reduce depression, stress, and anxiety [45] and is a predictor of quality of life and emotional distress [57]. Similarly, a negative association has been found between psychological distress and SOC whereas a positive relation has been observed between SOC and resilience [45,56–58]. Furthermore, SOC mediates the relationship between adverse experiences and positive well-being and plays a protective and mediating role between stressors and positive well-being [53,59,60]. The above evidence leads us to believe that SOC and resilience function as buffers and play a significant role in lowering frustration and stress and boosting TWB during the COVID-19 pandemic.

### 1.3. The Relation between SOC, Resilience, and Teacher Well-Being

SOC is a construct made up of three dimensions: comprehensibility, meaningfulness, and manageability [53,55]. Individuals with a higher SOC are better able to understand themselves and their social surroundings, reduce negative feelings, improve their overall physical health, reduce stress, and promote general well-being [46,59]. Furthermore, researchers discovered a substantiated, positive, and significant relationship between SOC and well-being, as well as a negative association with its negative outcomes [53,61–64]. Based on the preceding literature, this study regarded SOC as a critical strategy for reducing COVID-19 stress among teachers, as well as a mediator between COVID-19 stress and TWB. The purpose of this study was to investigate the roles of SOC and resilience as mediators between COVID-19 stress and TWB. It also looked at how COVID-19 stress, SOC, and resilience affected TWB.

Resilience is a potentially protective psychological resource that leads to long-term gains, allows recovery from life stressors, increases work and life satisfaction, builds social capital, aids in the acquisition of new knowledge and experiences, fosters better relationships with others, and promotes the search for a life purpose [47]. It also serves as a positive psychological mechanism that helps prevent harm, overcomes or compensates for risks [24], enables individuals to recover quickly and effectively from stressful experiences [45], and allows them to adjust to adversity in a favorable and positive manner [59]. Indeed, [24] created the resilience theory, which is applicable and necessary to everyday skills, all age groups, and all psychological situations. Meanwhile, [23,58] proposed the broaden-and-build theory of positive emotions, arguing that resilient individuals use positive emotions as core resources to rebound and find a purpose in life during stressful situations [60]. According to this model, resilience, as an intervention strategy that cultivates positive emotions, is more than just a method for healing and protecting oneself from pathology and distress [58,59].

In terms of the relationship between resilience and well-being, [45,61] contends that higher levels of resilience and optimism are strongly associated with higher levels of positive well-being or happiness and lower levels of stress. According to [13], the COVID-19 pandemic has impacted teachers' work-life balance and well-being. Furthermore, [61,62] discovered a strong interaction between stress, resilience, and well-being, and that resilience and low stress were important predictors of well-being. As a result, current research and policy would benefit from a study that investigates the relationship between resilience and positive well-being among teachers during the pandemic. Due to the fact of the urgency of the COVID-19 issue, this study examined the role of SOC and resilience [63–65] as well as the PERMA positive well-being model [31] in fostering TWB in order to investigate the direct and indirect effects of the core constructs during the pandemic. Additionally, the tools employed in this study were originally created for use in other cultural contexts. However, before performing mediation analysis in the current study, the measures were customized and validated for Ethiopian (African) culture. As a result, measurement invariance across socio-demographic characteristics and cross-cultural validation are also required [66–68].

Consequently, based on the latest scientific literature and the constructed theoretical framework in Figure 1, this study proposed the following hypotheses:

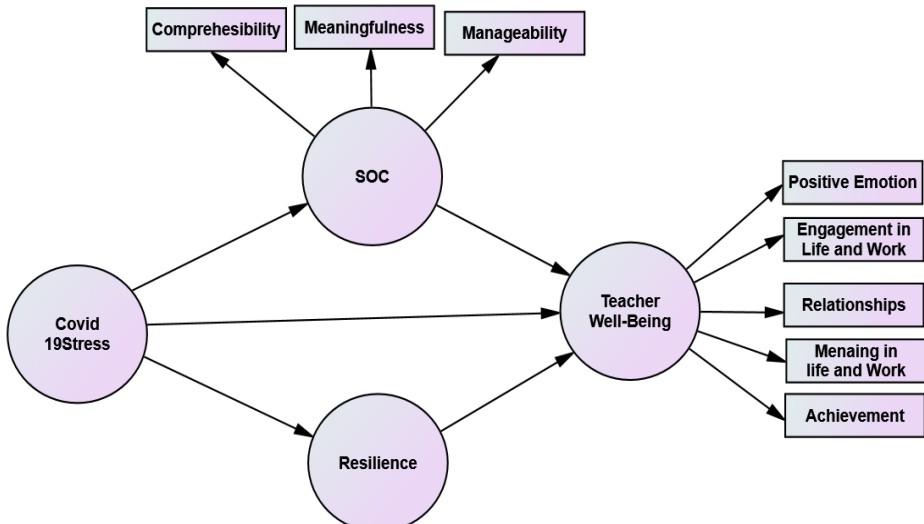

**Figure 1.** The relation between COVID-19 stress, resilience, sense of coherence, and teacher well-being.

**Hypothesis 1 (H1):** *COVID-19 stress would have a negative association with SOC, resilience, and TWB.*

**Hypothesis 2 (H2):** *Resilience and SOC as psychological resources and a lower level of COVID-19 stress would be linked with higher positive TWB.*

**Hypothesis 3 (H3):** *COVID-19 stress would directly and negatively influence SOC, resilience, and TWB.*

**Hypothesis 4 (H4):** *SOC and resilience would directly and positively affect TWB.*

**Hypothesis 5 (H5):** *SOC and resilience would mediate the relation between COVID-19 stress and TWB (see Figure 1).*

## 2. Materials and Methods

*2.1. Participants*

A cross-sectional design with an associational approach is employed to achieve the stated objectives. After the COVID restrictions were lifted, this study conducted paper-and-pencil tests and collected face-to-face data from 3–15 November 2020, as the universities in the study area have poor technological and Internet access. The samples were selected from public universities in the Amhara Regional State of Ethiopia. The study applied a stratified random sampling method to select the universities. Ethiopia is home to 50. Thus, the Ministry of Education grouped universities considered centers of excellence based on the establishment age, contribution to research and community service, the extent of international competition and collaboration, efficiency, infrastructure, and research publication [35,46,59]. Institutions in the study area were classified as research (Bahirdar and Gondar), applied (Wollo, Debre Berhan, and Debre Markos), and general (Debre Tabor, Mekidela amba Woldia, and Injibara) universities. Out of them, one from each classification was considered. Moreover, the samples obtained from these universities were representative of the country, because the Federal Ministry of Education assigned university teachers from different regions. For instance, a random sample of 883 teachers completed the questionnaires, but 47 were excluded before analysis because of incomplete data, and the response rate was 95%. Therefore, the remaining participants consisted of 630 male (75.4%) and 206 female (24.6%) public university teachers with a mean age of 32.81 years and an SD of 6.42. Specifically, 281 (33.6%), 370 (44.3%), and 185 (22.1%) teachers belong to Bahirdar University, Wollo University, and Debre Tabor University, respectively.

### 2.2. Instruments

The main tools included (a) demographic characteristics (gender, age, educational qualification, university, and experience in teaching), and (b) main measures to assess the variables of interest:

### 2.2.1. The Perceived Stress Scale of COVID-19

The Perceived Stress Scale of COVID-19 (PSS-10-C; [69]: The original PSS-10 was developed by [70] to measure how different situations influence an individual's feelings and perceived stress (see supplementary S1). The authors of [69] modified and adapted this instrument to the COVID-19 pandemic. The PSS-10-C is a unidimensional 10-item scale scored on a five-point Likert scale (never = 0, almost never = 1, occasionally = 2, almost always = 3, and always = 4). Items 1, 2, 3, 6, 9, and 10 were scored directly from 0 to 4, while items 4, 5, 7, and 8 were given the reverse (4 to 0) [69]. Scores ranged from 0 to 40, with cut-off points for high (25–40), moderate (14–24), and low (0–13) levels of perceived stress [69]. The construct showed high reliability in Colombian studies [69,71,72]. In this study, internal consistency for the PSS-10-C scale was acceptable with a Cronbach alpha and composite reliability where $\alpha = 0.97$; CR = 0.75 (see Table 1).

### 2.2.2. The Brief Resilience Scale (BRS)

The BRS, developed by [73], was used to measure the participants' resilience level, that is, teachers' ability to recover from adversity and stress. The scale consists of six items, three of which are positively worded and scored (1, 3, and 5), while the other three are negatively worded and scored (2, 4, and 6) (see supplementary S2). The participants rated each item from 1 (strongly disagree) to 5 (strongly agree). The BRS is a unidimensional factor, and the cut-off point above the average score indicates more resilience, while below the average implies low resilience [73]. Indeed, ref. [73] found that the psychometric properties of the BRS Amharci version (see Supplementary Materials) have excellent reliability. In this study, the reliability coefficient of BRS was acceptable with a Cronbach alpha of 0.86 and CR of 0.95 (see Table 1).

### 2.2.3. The Sense of Coherence Scale (SOC-13)

The SOC-13 is a measure developed by [74] consisting of 13 items with a seven-point Likert response format. It evaluates the participants' SOC. It contains three dimensions: manageability (four items: 3, 5, 10, and 13), comprehensibility (five items: 2, 6, 8, 9, and 11), and meaningfulness (four items: 1, 4, 7, and 12) [75], whose measurement depends on each item's context (see supplementary S3). Five items (1, 2, 3, 5, and 7) were reverse-scored, and the total score can range from 13 to 91; a higher score indicates a higher SOC [76]. In this study, the Cronbach's alpha for SOC—comprehensibility, manageability, and meaningfulness—were ($\alpha = 0.84$; CR = 0.88), ($\alpha = 0.90$; CR = 0.85), and ($\alpha = 084$; CR = 0.90), respectively, indicating the internal consistency for the scale (see Table 1).

### 2.2.4. The PERMA—Profiler Questionnaire

The PERMA—Profiler Questionnaire (PERMA; [77]: PERMA was designed to measure positive well-being dimensions [22]. It consists of 23 items, of which 15 measure the PERMA profile and 8 are filler items (see supplementary S4). However, this study used the 15-item measure to assess the PERMA pillars. Based on the item contents by [78], this study rated each item on a range from 0 (never, not at all, or terrible) to 10 (always, completely, or excellent, respectively). The scale has good model construct validity and reliability [22]. As shown in Table 3, The reliability coefficients of the five PERMA well-being dimensions were as follows (a) (positive emotion: $\alpha = 0.96$; CR = 0.96), (b) engagement: $\alpha = 0.964$; CR = 0.96; (c) positive relationships: $\alpha = 0.95$; CR = 0.95; (d) meaning in life: $\alpha = 0.94$; CR = 0.94; and (e) accomplishment: $\alpha = 0.95$; CR = 0.959, which implies that the scale had excellent internal consistency (see Table 1).

**Table 1.** Means, standard deviations, normal distributions, composite reliability (CR), Cronbach's alpha (α) values, and correlations (r) among the study constructs (*n* = 836).

| Variables | M | SD | Sk | Ku | CR | 1 | 2 | 3 | 4 | 5 | 6 | 7 | 8 | 9 | 10 | 11 | 12 |
|---|---|---|---|---|---|---|---|---|---|---|---|---|---|---|---|---|---|
| 1 | 14.37 | 4.889 | 1.35 | 1.50 | 0.96 | (0.961) | 0.739 ** | 0.781 ** | 0.821 ** | 0.819 ** | 0.929 ** | 0.392 ** | 0.343 ** | 0.196 ** | 0.431 ** | 0.275 ** | −0.236 ** |
| 2 | 14.09 | 4.589 | 1.21 | 2.01 | 0.96 | | (0.957) | 0.726 ** | 0.703 ** | 0.709 ** | 0.863 ** | 0.383 ** | 0.270 ** | 0.114 ** | 0.358 ** | 0.283 ** | −0.169 ** |
| 3 | 15.21 | 4.428 | 1.37 | 2.56 | 0.95 | | | (0.947) | 0.771 ** | 0.716 ** | 0.887 ** | 0.322 ** | 0.284 ** | 0.164 ** | 0.357 ** | 0.287 ** | −0.204 ** |
| 4 | 14.72 | 4.622 | 1.14 | 1.33 | 0.94 | | | | (0.935) | 0.787 ** | 0.909 ** | 0.382 ** | 0.341 ** | 0.182 ** | 0.418 ** | 0.268 ** | −0.270 ** |
| 5 | 14.07 | 4.667 | 1.03 | 1.03 | 0.95 | | | | | (0.951) | 0.899 ** | 0.386 ** | 0.316 ** | 0.141 ** | 0.391 ** | 0.290 ** | −0.234 ** |
| 6 | 72.50 | 20.83 | 1.65 | 1.93 | 0.95 | | | | | | (0.974) | 0.416 ** | 0.347 ** | 0.178 ** | 0.436 ** | 0.312 ** | −0.248 ** |
| 7 | 15.43 | 4.385 | 0.182 | −0.958 | 0.85 | | | | | | | (0.901) | 0.361 ** | 0.293 ** | 0.795 ** | 0.201 ** | −0.132 ** |
| 8 | 18.01 | 3.457 | 0.104 | −0.109 | 0.90 | | | | | | | | (0.841) | 0.166 ** | 0.665 ** | 0.242 ** | −0.147 ** |
| 9 | 22.76 | 4.057 | −0.160 | −0.646 | 0.88 | | | | | | | | | (0.878) | 0.689 ** | −0.009 | −0.029 |
| 10 | 56.20 | 8.58 | −0.561 | −0.036 | 0.88 | | | | | | | | | | (0.855) | 0.196 ** | −0.140 ** |
| 11 | 26.67 | 3.816 | −0.506 | 0.380 | 0.95 | | | | | | | | | | | (0.858) | −0.152 ** |
| 12 | 22.86 | 10.14 | −1.19 | 0.922 | 0.75 | | | | | | | | | | | | (0.967) |

Notes: ** *p* < 0.001 (two-tailed); Cronbach's alpha (**α**) in **diagonal bold**, SD = standard deviation, 1 = positive emotion, 2 = engagement, 3 = relation, 4 = meaning in life, 5 = achievement, 6 = PERMA well-being model, 7 = manageability, 8 = meaningfulness, 9 = comprehensibility, 10 = sense of coherence, 11 = resilience, 12 = COVID-19 stress.

### 2.3. Procedures of the Study

The questions were filled out by the participants using paper and pencil. The data-collecting process was carried out in accordance with the American Psychological Association's standards as well as the Helsinki Declaration of 1964, 21 CFR 50 (Protection of Human Subjects), and 21 CFR 56 (Institutional Review Boards). The study subsequently received an ethical approval letter and research clearance from the Internal Review Board (IRB) of the University of Szeged (certificate number: Ref. 26). Before taking part in the trial, each subject provided their informed consent. The original versions of the four scales were then translated into Amharic using a combination of backward and forward translation techniques. Three professors with years of research and translation experience, one from the university's English Language Improvement Training Center, one from clinical psychology, and one from developmental psychology, conducted the Amharic translation. There were no errors in the translation. Finally, the participants we had were given the questionnaires. The method of data analysis is described in the next step.

### 2.4. Data Analysis

Versions 26 of both Analysis of Moment Structures (AMOS) and the Statistical Package for the Social Sciences (IBM Corporation) were used in the study. We addressed multicollinearity before beginning the primary data analysis process by determining the correlation between the variable values, which should be greater than 0.90, and by examining the normality of distributions in accordance with [79,80] advice. We tested the normality of distributions in the following step. Skewness and kurtosis values fall within the range of [−2] and [+2], which is sufficient to demonstrate the data's normal distribution [81]. The data were analyzed after the necessary conditions were satisfied.

Additionally, Cronbach's alpha coefficient ($\alpha$) and composite reliability (CR) are used to evaluate the reliability scores of the current study. According to [81], the reliability value $\geq 0.9$ = excellent; the value ranges from $\alpha$ 0.9 to 0.8 = good; 0.8 to 0.7 = acceptable; 0.7 to 0.6 = questionable; $\alpha$ 0.6 to 0.5 = poor, and $0.5 > \alpha$ = unacceptable [78,81]. After checking the reliability, convergent, and discriminant validity, we examined the measurement model.

In addition, the study used confirmatory factor analysis based on the recommendation of [82–84] to check the psychometric properties of the scales of each construct's measurement. In order to confirm that the measures are equivalent, this study carried out four stages of measurement invariance on the four core components (COVID-19 Stress, resilience, TWB, and SOC). The relations of the constructs were also carried out using Pearson correlation. Then, we tested the structural or proposed mediation model using the bootstrap method. More details drawn upon the structural model were based on previous research theory. Finally, structural Equation Modelling (SEM) analysis, including measurement and structural models' tests, was performed.

The structural model draws upon theory, literature, and research objectives to differentiate which predictor variables explain each criterion variable. In contrast, the measurement model was used to measure all variables merged to represent the theory [84]. Hence, the measurement model, including COVID-19 stress, resilience, sense of coherence (SOC), and teacher well-being (TWB) constructs, was formulated. The cut-off values for acceptable fitness of the indices of the structural equation modeling (SEM) were $\chi^2$ = insignificant and GFI, RFI, TLI, and CFI $\geq 0.90$ [79,80]; this study considered SRMR and RMSEA $\leq 0.10$ as its criteria [85]. However, in large data samples, the $\chi^2$ test is extremely sensitive and will show a probability to be significant, and it is not advisable to draw an absolute cut-off value for RMSEA [84]. Before conducting SEM, we conducted a CFA analysis of the scales as recommended by [82–84].

We used the ML approach, a typical estimate-based SEM, to analyze the hypothetical model shown in Figures 1 and 2. To check for measurement issues, the problem of Common-Method Biases (CMB) was conducted. Common-method biases can have an impact on social scientific research, particularly those using paper-and-pencil instruments. Examples of these influences include the test's content, the format of the response, the general

instructions for the items, and the purpose of the subject's participation [86,87]. In this study, the following actions were taken to resolve such problems: (a) Prior to the instrument's administration, subject-matter experts assessed the accuracy of each item's content or face. (a) All participants gave their consent after being fully informed, and their names were coded anonymously. (c) Some items received reversed scores. (d) Different sources and cultural settings were used to create the predictor and criteria variables. (e) Factor variance was computed to take measurement mistakes into account. According to Harman's single-factor test standards, the common-method bias test was conducted [86,87]. No substantial common-method biases with computed variances below the cutoff of 50% were found in this study (36.6 percent).

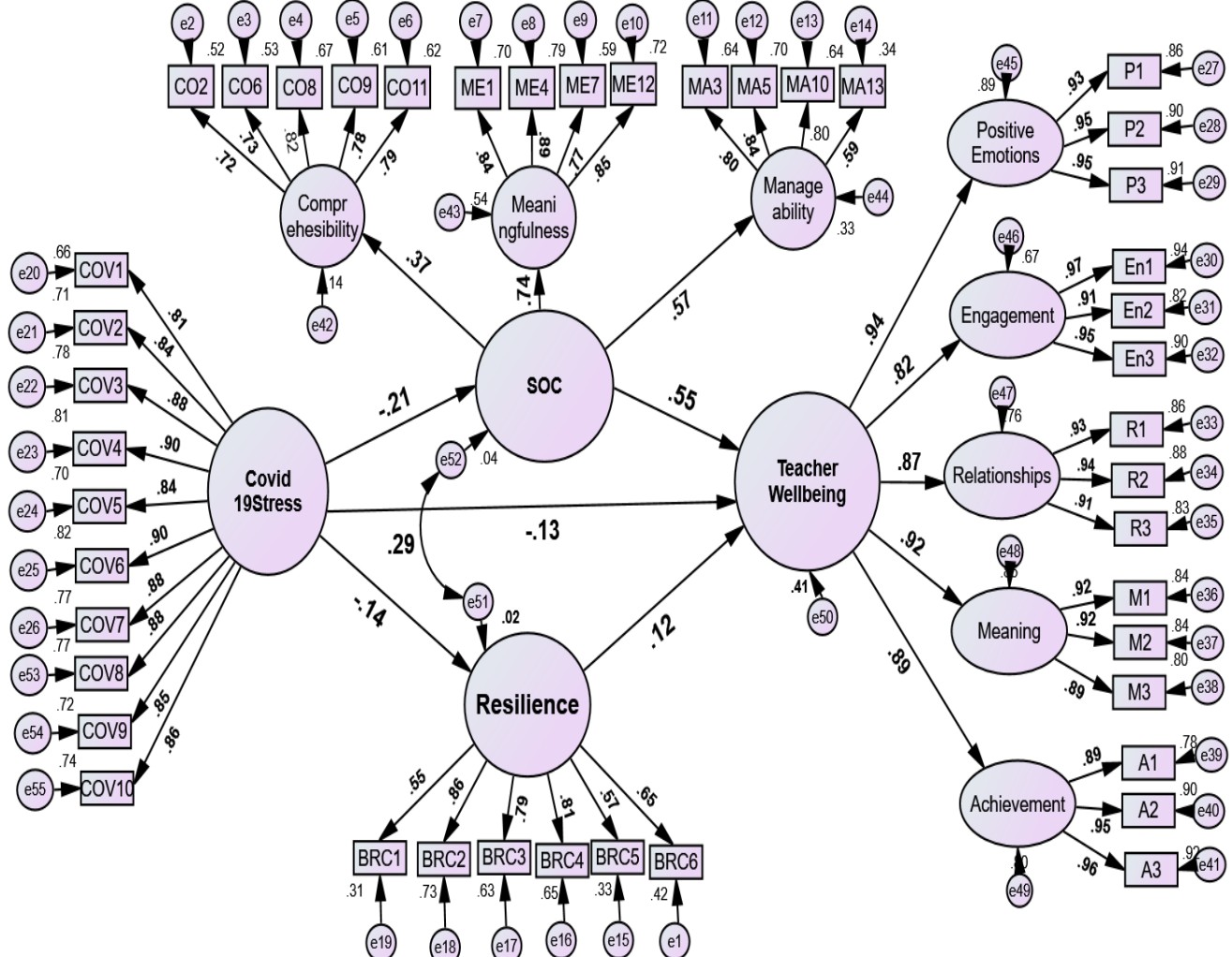

**Figure 2.** Mediation model: the mediating role of sense of coherence and resilience on teachers' well-being. Comprehensibility = CO2, CO6, CO8, CO11; Meaningfulness = ME1, ME4, ME7, ME12; Manageability = MA3, MA5, MA10, MA13; Positive emotion = P1–P3; enagemenemnt = En1–En3; Relationships = R1–R3; Meaning = M1–M3'. Acheievemnt = A1–A3; resileince = BRC1–BRC6; COVID−19 Stress = COV1–COV10.

Overall, there were several key justifications for employing SEM in this study: (1) Using a variety of techniques, this study examines the relationships between latent constructs [82,84,88]; (2) Confirming the factor structure of a psychological instrument is advised [84,89]. (3) The complexity of our suggested model stems from its examination of structural factor models (CFA), direct and indirect (mediated) effects, and other intricate

relationships between variables. (4) This study employs bootstrap techniques for inferences about indirect mediation.

## 3. Results

### 3.1. Preliminary Analysis

3.1.1. Descriptive Statistics, Bivariate Correlation, and Primary Construct's Reliability

Table 1 displays values for internal consistency, descriptive statistics (mean and standard deviation), distribution normality using kurtosis and skewness, and correlations for all of the main constructs. A skewness of 2 or a kurtosis of 4 indicates that the data is normally distributed [79,80]. Skewness values in this study ranged from 0.104 to 1.65, and kurtosis scores ranged from 0.036 to 2.0, indicating that all constructs had a relatively normal distribution (see Table 1).

Following that, multi-collinearity was investigated using Tolerance and Variance Inflation Factor (VIF) values. The absence of multi-collinearity or singularity is determined by inspecting the determinant of covariance matrices; checking the correlations among the construct values, which must be less than 0.90; using tolerance and VIF; and evaluating the normality assumption. The study also used [82,83] to investigate construct outliers. Furthermore, if the Tolerance values for each predictor variable in the model are close to one, there is no problem with multi-collinearity, and the opposite is true when the Tolerance value is close to zero [84]. The VIF statistic should be in the range of zero (0) to five (5), with a lower value (closer to zero) preferred. If the VIF statistic is greater than five (5), the data is highly correlated and there is a multi-collinearity problem [84]. High VIF values indicate that a given predictor variable is a linear combination of the others [84]. As a result, the VIF for this study was less than 5, and the Tolerance of all independent variables was greater than or equal to 0.01. Therefore, both VIF and Tolerance confirmed that the independent variables are free of multi-collinearity problems (see Table 2).

**Table 2.** Multi-collinearity statistics of Tolerance and VIF [a].

| Term | Uncoeff | Stancoff | t-Value | *p*-Value | Tolerance | VIF |
|---|---|---|---|---|---|---|
| COVID-19 Stress | −0.336 | −0.163 | −5.415 | 0.001 | 0.964 | 1.037 |
| SOC | 0.902 | 0.371 | 12.215 | 0.001 | 0.949 | 1.054 |
| Resilience | 1.173 | 0.215 | 7.054 | 0.001 | 0.946 | 1.057 |

[a] Dependent Variable: teacher well-being.

A correlation matrix was performed and presented for the first hypothesis. The findings supported hypothesis 1 by confirming a significant negative correlation between COVID-19 Stress and SOC (r = 0.140, *p* < 0.01), resilience (r = 0.204, *p* < 0.01), and TWB (r = 0.234, *p* < 0.01). Additionally, there was a good and substantial correlation between TWB, SOC, and resilience (see Table 1).

### 3.1.2. Convergent and Discriminant Validity

The researchers evaluated the validity of SOC and TWB, resilience, and COVID-19 Stress based on their respective AVE and MSV scores (see Table 3). This study discovered that all constructs (TWB, SOC, resilience, and COVID-19) have good convergent validity (AVE > 0.05), implying that the corresponding items are made up of core factors with acceptable correlation (see Table 3). We used an AVE greater than MSV method to test the discriminant validity of the four main constructs, because each item explains the latent constructs in each factor.

**Table 3.** Convergent and Discriminant Validity Indices of the Main Constructs.

| | Convergent and Discriminant Validity | | |
|---|---|---|---|
| **No** | **Models** | **AVE (>0.50 \*)** | **MSV** |
| | **Teacher Well-Being (TWB)** | | |
| 1 | Positive emotion | 0.89 | 0.75 |
| 2 | Engagement | 0.88 | 0.58 |
| 3 | Relationships | 0.86 | 0.67 |
| 4 | Meaning | 0.83 | 0.75 |
| 5 | Accomplishments | 0.87 | 0.73 |
| | **Sense of Coherence (SOC)** | | |
| 1 | Comprehensibility | 0.59 | 0.11 |
| 2 | Manageability | 0.58 | 0.11 |
| 3 | Meaningfulness | 0.70 | 0.16 |
| | COVID-19 Stress \*\* | 0.97 | |
| | Resilience \*\* | 0.75 | |

Note: Asterisk (\*) indicates a global rule of thumb of an acceptable level of validity based on the recommendation of Hair et al. (2019) [84] and Kline (2016) [79]; Asterisk (\*\*) indicates the constructs are unidimensional.

Because the constructs' AVE values were greater than MSV, this study also tested them (see Table 3). As a result, the TWB scale sub-constructs AVE were found to be greater than MSV, with the following results: (a) positive emotion (AVE = 0.89 > MSV = 0.75), (b) engagement (AVE = 0.88 > MSV = 0.58), (c) relationships (AVE = 0.86 > MSV = 0.67), (d) meaning (AVE = 0.83 > MSV = 0.75), and (e) achievement (AVE = 0.87 > MSV = 0.75). Furthermore, the SOC construct results for the sub-scales are as follows: (a) comprehensibility (AVE = 0.58 > MSV = 0.11), (b) manageability (AVE = 0.70 > MSV = 0.16), and (c) meaningfulness (AVE = 0.59 > MSV = 0.11). Finally, the AVE values for COVID-19 Stress and Resilience were both greater than 0.5 (AVE = 0.75). As a result, we can confidently conclude that the four primary constructs in Ethiopian higher education settings meet the convergent and discriminant validity requirements. Furthermore, the results show that the instruments are psychometrically sound.

### 3.1.3. Measurement Invariance (MI) of the Study Variables

Measurement invariance or equivalence (MI) refers to the unbiased measurement between two languages and cultural backgrounds using the same instrument [35,46,59,78], and it is needed to confirm comparative groups (culture, gender, age, education, etc.) [46,78]. Such differences are detected by applying MI across various group stages [66–68]. The researchers followed well-established scientific procedures using the four MI stages [66,67]. In stage 1, a configural invariance was conducted to establish a baseline model across groups without restriction, where the tested construct was the same across all groups [66–68]. In stage 2, we examined the metric measurement invariance (MMI) and the same constrained factorial loadings to the different groups that responded similarly to indicators. In stage 3, scalar measurement invariance or strong invariance (SMI) was performed. In this test, the indicator intercepts and the factor loadings were constrained similarly across groups. Finally, the residual measurement invariance or the strict invariance (RMI) was tested. It refers to the similarity of item residuals of metric and scalar invariant items [59,66,67]. The MI of the present study four-sequential-staged analysis used single and multi-group CFA following [66–68], and arrived at the following recommendation criteria: ΔTLI, 0 = perfect and ≤0.01 = acceptable, ΔRMSEA, 0.015 for metric, scalar, and residual invariance [46,66,67,78]. Consequently, in this research, university teachers' gender, university type (research, applied, and general universities), and teaching experience (<5, 6–10, and ≥11 years) on TWB, SOC, resilience, and COVID-19 stress showed an excellent fit to the data (see Table 4). In addition, the strict model (residual) was achieved, and all item loadings, intercepts, and residual variances were equivalent or equal across the three levels of experience in teaching.

**Table 4.** Fit Indices for the Configural, Metric, Scalar, and Residual Models Across Socio-demographic factors.

| Scales | Groups | Configural | | | Metric | | | Scalar | | | Residual | | |
|---|---|---|---|---|---|---|---|---|---|---|---|---|---|
| | | TLI | CFI | RMSEA | TLI | CFI | RMSEA | TLI | CFI | RMSEA | TLI | CFI | RMSEA |
| **COVID-119 Stress** | Gender | 0.932 | 0.947 | 0.091 | 0.940 | 0.947 | 0.086 | 0.947 | 0.948 | 0.080 | 0.951 | 0.946 | 0.078 |
| | University Type | 0.911 | 0.931 | 0.086 | 0.923 | 0.930 | 0.080 | 0.932 | 0.928 | 0.076 | 0.929 | 0.915 | 0.077 |
| | Experience | 0.909 | 0.921 | 0.086 | 0.913 | 0.920 | 0.085 | 0.921 | 0.916 | 0.081 | 0.915 | 0.905 | 0.084 |
| **Resilience** | Gender | 0.929 | 0.957 | 0.079 | 0.931 | 0.949 | 0.078 | 0.943 | 0.945 | 0.071 | 0.947 | 0.938 | 0.068 |
| | University Type | 0.931 | 0.958 | 0.065 | 0.928 | 0.943 | 0.069 | 0.927 | 0.920 | 0.067 | 0.920 | 0.909 | 0.064 |
| | Experience | 0.911 | 0.937 | 0.078 | 0.911 | 0.928 | 0.077 | 0.904 | 0.915 | 0.075 | 0.901 | 0.907 | 0.071 |
| **SOC** | Gender | 0.970 | 0.976 | 0.037 | 0.973 | 0.977 | 0.035 | 0.975 | 0.976 | 0.030 | 0.971 | 0.971 | 0.036 |
| | University Type | 0.952 | 0.962 | 0.038 | 0.951 | 0.957 | 0.039 | 0.954 | 0.954 | 0.037 | 0.951 | 0.947 | 0.039 |
| | Experience | 0.964 | 0.972 | 0.032 | 0.955 | 0.960 | 0.036 | 0.948 | 0.949 | 0.039 | 0.929 | 0.924 | 0.046 |
| **TWB** | Gender | 0.985 | 0.989 | 0.037 | 0.986 | 0.989 | 0.036 | 0.987 | 0.989 | 0.035 | 0.985 | 0.985 | 0.038 |
| | University Type | 0.978 | 0.983 | 0.037 | 0.979 | 0.983 | 0.036 | 0.973 | 0.975 | 0.041 | 0.969 | 0.968 | 0.044 |
| | Experience | 0.980 | 0.985 | 0.033 | 0.980 | 0.983 | 0.034 | 0.966 | 0.968 | 0.044 | 0.944 | 0.946 | 0.057 |

Notes. N = 836, $p < 0.001$, RMSEA = root mean squared error of approximation, TLI = Tucker-Lewis index; CFI = comparative fit index.

### 3.1.4. Measurement Model

AMOS statistical software was used to perform the SEM statistical analysis, which included measurement and structural model tests based on the recoomendation of [82–84,88,89]. A confirmatory factor analysis (CFA) was performed as an initial stage to determine whether the measurement model provides an acceptable fit to the data. The measurement model was then developed, with latent constructs for COVID-19 Stress, resilience, sense of coherence (SOC), and teacher well-being (TWB) included separately and together. Since then, TWB observed variables have included positive emotion (P), engagement (E), relationships (R), meaning (M), and accomplishments (A). Similarly, the SOC subscales of Comprehensibility (CO), Manageability (MA), and Meaningfulness (ME) were defined as three observed variables. COVID-19 stress and resilience, on the other hand, are one-dimensional constructs. As a result, we relied on the most commonly used goodness of fit statistics in this study: the goodness of fit index (GFI), the Relative Fit Index (RFI), the Tucker-Lewis index (TLI), the comparative fit index (CFI), the root mean square error of approximation (RMSEA), and the Standardized Root Mean Square Residual (SRMR). Poor fit > 0.85, mediocre fit = 0.85–0.90, acceptable fit = 0.90–0.95, close fit = 0.95–0.99, and exact fit = 1.00 are the recommended cut-points for GFI, RFI, TLI, and CFI [83]. Poor fit = greater than 0.10, mediocre fit = 0.08 to 0.10, good fit = 0.05 to 0.08, close fit = 0.01 to 0.05, and exact fit = 0.00 for RMSEA and SRMR [85]. In this study, for example, the measurement model of the constructs using the maximum likelihood method produced a good fit to the data and was presented as follows. Meanwhile, the construct validity of the Ethiopian Amharic version of the COVID-19 PSS-10-C Perceived Stress Scale was tested using the CFA model, and the goodness-of-fit values are as follows: 2(35) = 443.37, $p < 0.001$, GFI = 0.900, RFI = 0.938, TLI = 0.943, CFI = 0.955, SRMR = 0.035, and RMSEA = 0.118 (0.109–0.128. Second, the construct validity of the SOC-13 was confirmed in this study using a robust maximum likelihood (ML) estimation method: 2 (60) = 188.20, $p< 0.001$, GFI = 0.966, RFI = 0.960, TLI = 0.973, CFI = 0.978, SRMR = 0.035, and RMSEA = 0.049 (0.041–0.065) (see Table 5). Additionally, the CFA noted on the same Table that the BRS measurement model suited the data well: 2(9) = 110.97, $p < 0.01$, GFI = 0.955, RFI = 0.0916, TLI = 0.922, CFI = 0.953, SRMR = 0.036, and RMSEA = 0.116 (0.098–0.136).The TWB model, which was estimated using the ML method, confirmed the best model fit: 2(80) = 266.59, GFI = 0.960, RFI = 0.980, TLI = 0.985, CFI = 0.990, SRMR = 0.039, and RMSEA = 0.053 (0.046–0.060) (see Table 5). Finally, the goodness-of-fit measurement models for all four variables were acceptable:

2(846) = 2060, $p$ < 0.001, GFI = 0.900, RFI = 0.938, TLI = 0.964, CFI = 0.966, SRMR = 0.039, and RMSEA = 0.040 (0.038 to 0.042). By [85]'s cut-off points, the data had an acceptable fit and met the recommended criteria of the CFA models, with GFI, RFI, TLI, and CFI 0.90 and RMSEA and SRMR 0.10.

**Table 5.** Confirmatory Factor Analysis of the Constructs: the Structural Model and Measurement Model (N = 836).

| Fitness of Indices | Confirmatory Factorial Analysis of the Constructs | | | | | | | | |
|---|---|---|---|---|---|---|---|---|---|
| | $\chi^2$ | df | $p$-Value | GFI | RFI | TLI | CFI | SRMR | RMSEA |
| **COVID 19 Stress** | 443.37 | 35 | 0.001 | 0.900 | 0.938 | 0.943 | 0.955 | 0.035 | 0.118(0.109 to 0.128) |
| **SOC** | 188.20 | 60 | 0.001 | 0.966 | 0.960 | 0.973 | 0.978 | 0.040 | 0.049 (0.041 to 0.058) |
| **Resilience** | 110.97 | 9 | 0.001 | 0.955 | 0.916 | 0.922 | 0.953 | 0.028 | 0.116 (0.098 to 0.136) |
| **TWB** | 266.59 | 80 | 0.001 | 0.960 | 0.980 | 0.985 | 0.990 | 0.039 | 0.053 (0.046 to 0.060) |
| **Rule of thumb** | | | | ≥0.90 | ≥0.90 | ≥0.90 | ≥0.90 | ≥0.10 | ≥0.08 |

Note: GFI = goodness of fit index; RFI = relative non-centrality index; TLI = Tucker-Lewis index; CFI = comparative fit index, SRMR = standardized root mean square residual; RMSEA = root mean squared error of approximation.

### 3.1.5. Structural Model

In this study, the structural model of the mediation model using the maximum likelihood method did produce a good model fit to the data, $\chi^2$ (888) = 2060, $p$ < 0.001, GFI = 0.901, $\chi^2$/df = 2.32, RFI = 0.936, TLI = 0.964, CFI = 0.967, SRMR = 0.039, and RMSEA = 0.040 (0.038 to 0.042) (see Table 6). This implies that our meditation model has acceptable structural validity, as supported by [84,85]'s cut-off points.

**Table 6.** Models of Goodness-of-fit indices.

| Path Model | Types of Models | $\chi^2$ (df) * | $\chi^2$/df | GFI | RFI | TLI | CFI | SRMR | RMSEA | Rule of Thumb |
|---|---|---|---|---|---|---|---|---|---|---|
| **Model 1** | Structural | 1604(654) * | 2.45 | 0.908 | 0.947 | 0.968 | 0.970 | 0.035 | 0.042 | SRMR and RMSEA ≤ 0.8 |
| **Model 2** | Structural | 1273(426) * | 2.98 | 0.909 | 0.952 | 0.967 | 0.970 | 0.036 | 0.049 | $\chi^2$/df < 5, GFI, RFI, TLI, |
| **Model 3** | Measurement | 2060(888) * | 2.32 | 0.901 | 0.938 | 0.964 | 0.967 | 0.039 | 0.038 | CFI ≥ 0.90 |
| | Structural | 2060(888) * | 2.32 | 0.901 | 0.938 | 0.964 | 0.967 | 0.039 | 0.038 | |

Note: * $p$ < 0.001, $\chi^2$ = chi-squared, df = degrees of freedom, GFI = goodness of fit index, TLI = Tucker Lewis index, CFI = comparative fit index, SRMR = standardized root means square residual, RMSEA = root mean error square of approximation. **Model 1**: COVID-19 Stress → SOC → Teachers' well-being. **Model 2**: COVID-19 Stress → Resilience → Teachers' well-being. **Model 3**: COVID-19 Stress → SOC and Resilience → Teachers' well-being (whole structural mediated model; see Figure 2).

### 3.2. Status of the Primary Constructs

The second hypothesis was that resilience and SOC as psychological resources and a lower level of COVID-19 stress would be associated with higher positive TWB. Based on the suggested cut-off points, this study computed the mean scores of the variables. The highest scores were for resilience (26.7 out of 30, SD = 3.81) and SOC (56.2 out of 91, SD = 8.60), followed by TWB (72.5 out of 150, SD = 20.83). Hence, Hypothesis 2 was confirmed. The COVID-19 stress scores ranged from 0 to 40, with cut-off points for high (25–40), moderate (14–24), and low (0–13) levels of perceived stress [69]. Using these suggested cut-off points, the mean score of the COVID-19 stress construct (mean = 22.86, SD = 10.14) was found to be moderate. These findings revealed that higher resilience and SOC scores and a moderate TWB level from the PERMA model played significant roles in the participants' resistance to COVID-19 stress.

### 3.3. Mediation Analysis

Through SOC and resilience, as shown in Tables 6 and 7, this study explored COVID-19 stress and TWB. The direct and indirect impacts of the predictors factors on the criterion variables were examined and reported (see Figure 2 and Table 7). The results showed

a negative and significant standardized direct effect path from COVID-19 stress to SOC (β = −0.21, [BC 95% bootstrap CI: −0.275 to −0.124], $p < 0.01$), while COVID-19 stress also had a significant and negative direct effect on resilience (β = −0.141 [95% bootstrap CI: −0.212 to −0.074], $p < 0.01$) and TWB (β = −0.132, 95% bootstrap CI: −0.183 to −0.074, $p < 0.01$). These data are consistent with Hypothesis 3.

**Table 7.** Bootstrapping standardized direct and indirect effect using 95% biased corrected confidence interval predicting teachers' well-being and the structural model fitness of indices (N = 836).

| Path Model | | Bootstrap 95% CI | | | |
|---|---|---|---|---|---|
| | | Beta | LBC | UBC | *p*-Value |
| **Standardized Direct Effect** | | | | | |
| **Predictors** | **Outcome Variables** | | | | |
| COVID-19 stress | SOC | −0.205 | −0.275 | −0.124 | 0.001 |
| COVID-19 stress | Resileince | −0.141 | −0.212 | −0.074 | 0.001 |
| COVID-19 stress | TWB | −0.132 | −0.183 | −0.074 | 0.001 |
| SOC | TWB | 0.554 | 0.488 | 0.629 | 0.001 |
| Resilience | TWB | 0.120 | 0.047 | 0.171 | 0.01 |
| **Standardized Indirect Effect** | | | | | |
| COVID-19 Stress → SOC and Resilience → | TWB (Figure 2) | −0.130 | −0.180 | −0.083 | 0.001 |
| COVID-19 Stress → SOC → | TWB (Figure 3) | −0.120 | −0.167 | −0.071 | 0.001 |
| COVID-19 Stress → Resilience → | TWB (Figure 4) | −0.039 | −0.060 | −0.021 | 0.001 |

Note: CI = confidence interval, LBC = lower bound, UBC = upper bound, SOC = sense of coherence, TWB = teacher well-being.

The results also showed a significant and positive direct effect of SOC and resilience on TWB (β = 0.554, 95% bootstrap CI: 0.490 to 0.630, $p < 0.01$, and β = 0.120 [95% bootstrap CI: 0.047 to 0.171], $p < 0.01$, respectively), confirming Hypothesis 4.

Moreover, the indirect effect of COVID-19 stress (see Table 4) on TWB mediated through SOC and resilience was significant (β = −0.130, 95% bootstrap CI: −0.180 to −0.083), supporting Hypothesis 5.

The next step is to test using TWB as the dependent variable, COVID-19 stress as the predictor variable, and SOC as the partial mediator variable. This study found that COVID–19 stress had a significant and negative direct effect on TWB (β = −0.144, 95% bootstrap CI: −0.198 to −0.084, $p < 0.01$). Meanwhile, the direct effect of SOC on TWB was positive and significant (β = 0.590, 95% bootstrap CI: 0.520 to 0.650, $p < 0.001$).

The indirect effect of COVID-19 stress on TWB through SOC was significant as well (β = −0.120, 95% bootstrap CI: −0.167 to −0.071, $p < 0.01$), which confirms the partial medation. The mediation model through SOC indicated an acceptable fit: $\chi^2$ (654) = 1604, $p < 0.001$, $\chi^2/df$ = 2.45, GFI = 0.909, RFI = 0.947, TLI = 0.968, CFI = 0.970, SRMR = 0.035, and RMSEA = 0.042 (0.039 to 0.044) (see Table 6). This showed that the structural validity of the model was acceptable, which is also supported by [85]'s cut-off points.

COVID-19 stress produced a negative and significant indirect effect on TWB through resilience (β = −0.039, 95% bootstrap CI: −0.060 to −0.021, $p < 0.001$), which supported the partial mediation role of resilience. This study also observed a significant and negative direct effect of COVID-19 stress on TWB through resilience as a mediator (β = −0.223, 95% bootstrap CI: −0.268 to −0.174, $p < 0.01$), followed by COVID-19 stress on resilience (β = −0.141, 95% bootstrap CI: −0.212 to −0.074, $p < 0.01$).

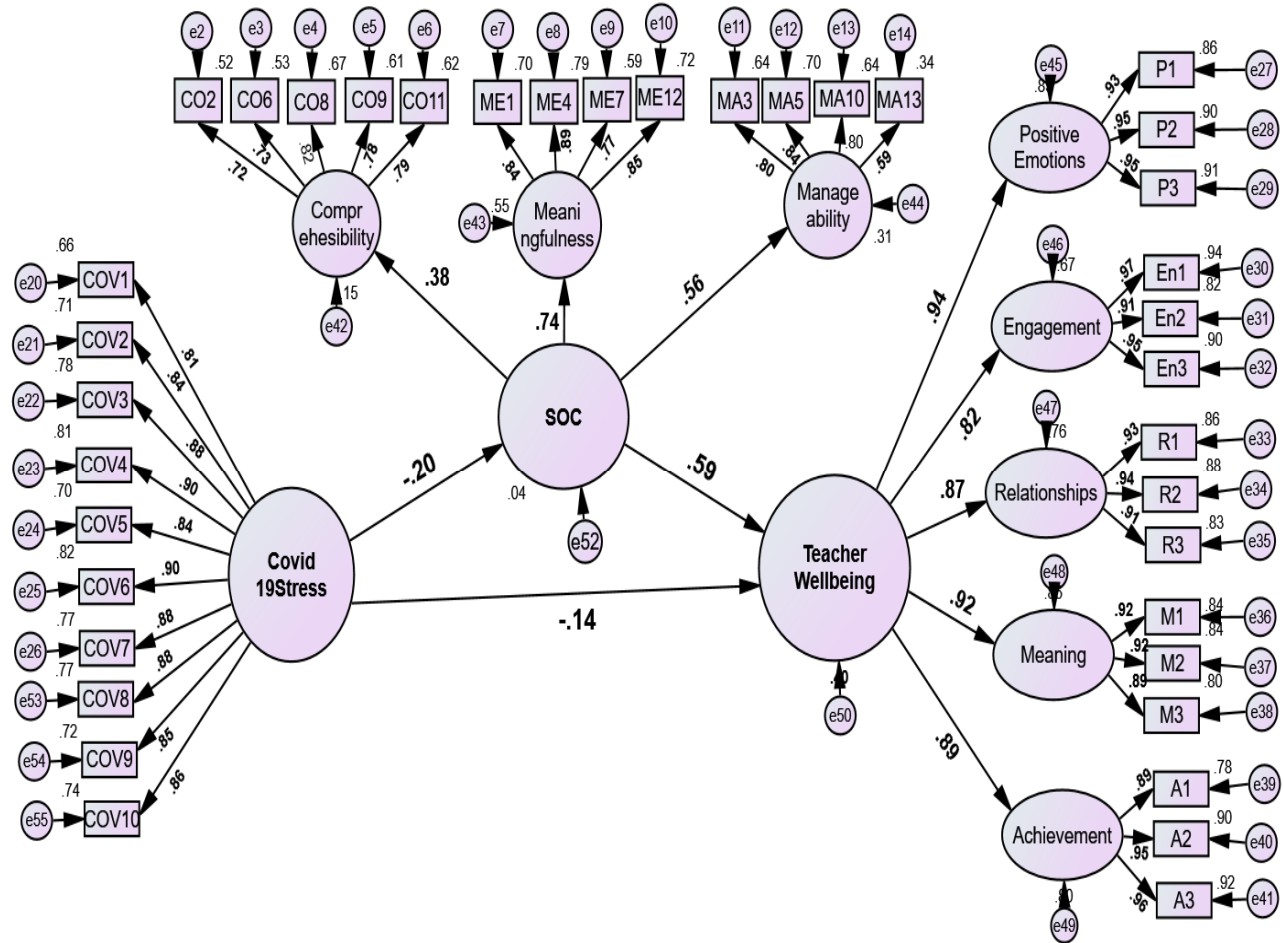

**Figure 3.** Mediation model: the mediating role of sense of coherence between COVID-19 Stress and teachers' well-being. Comprehensibility = CO2, CO6, CO8, CO11; Meaningfulness = ME1, ME4, ME7, ME12; Manageability = MA3, MA5, MA10, MA13; Positive emotion = P1–P3; enagemenemnt = En1–En3; Relationships = R1–R3; Meaning = M1–M3. Acheievemnt = A1–A3; resileince, COVID-19 Stress = COV1–COV10.

The direct effect of resilience on TWB was also positive and significant (β = 0.280, 95% bootstrap CI: 0.240 to 0.321, *p* < 0.001). The model showed decent goodness-of-fit index values—$\chi^2$ (426) = 1273.24, *p* < 0.001, $\chi^2/df$ = 2.98, GFI = 0.909, RFI = 0.952, TLI = 0.967, CFI = 0.970, SRMR = 0.057, RMSEA = 0.049 (0.046 to 0.052)—which generated tangible evidence to the partial mediation role of resilience. GFI, RFI, TLI, and CFI values of 0.90 or more indicate a model's good fit. This study also recommended structural model fit values to test the mediation effects as supported by [85], that the GFI, TLI, and CFI values of 0.90 indicate acceptable fit whereas values of 0.95 and above indicate good fit.

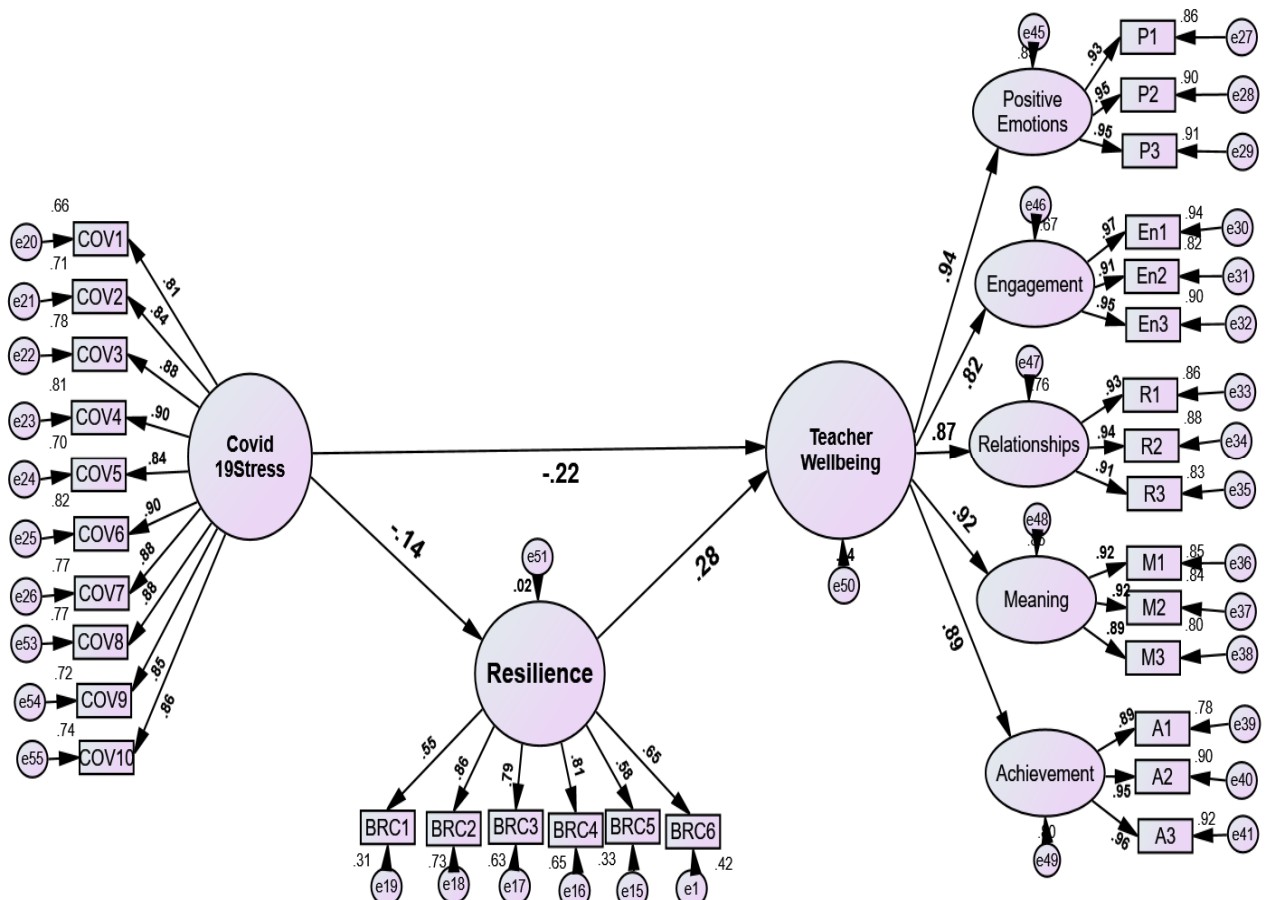

**Figure 4.** Mediation model: the mediating role of resilience between COVID-19 stress and teachers' well-being. Positive emotion = P1–P3; engagement = En1–En3, Relationships = R1–R3. Meaning = M1–M3'. Acheievemnt = A1–A3; resilience = BRC1–BRC6; COVID-19 Stress = COV1–COV10.

## 4. Discussion

This study validates the COVID-19 Perceived Stress Scale, the Sense of Coherence Scale, the Brief Resilience Scale, and the PERMA Profiler Questionnaire. This study verified the measuring model's convergent and discriminant validity to ensure validity. Prior to evaluating the mediation model, Cronbach alpha and composite reliability were used to assess the reliability of each of the four key constructs. The normality distribution, inter-item correlation, Cronbach alpha, and composite reliability of the COVID-19 Perceived Stress Scale, Sense of Coherence Scale, Brief Resilience Scale, and PERMA Profiler Questionnaire, as well as construct validity, convergent, discriminant, and measurement invariances, are investigated in Ethiopian Amharic. As a preliminary analysis, the measurements must be cross-culturally confirmed using a self-reported instrument. A number of scientific analyses were used to evaluate the COVID-19 Perceived Stress Scale, the Sense of Coherence Scale, the Brief Resilience Scale, and the PERMA Profiler Questionnaire in Ethiopian Amharic. This investigation validated the tools' good psychometric properties. These measures are used internationally and across various cultures. The construct validity of the COVID-19 Perceived Stress Scale, Sense of Coherence Scale, Brief Resilience Scale, and PERMA-Profiler Questionnaire was investigated using CFA analysis. Furthermore, the structural model used in this study looked at how COVID-19 stress affected TWB both directly and indirectly via SOC and resilience. To the best of the authors' knowledge, no findings in the field of education, particularly with regard to university teachers, have been found. The originality stems from its assessment of the possible role of SOC and resilience in promoting positive well-being and lowering COVID-19 stress.

Indeed, it has become relevant in today's higher education to conduct a study on the contemporary global issue of COVID-19 and its association with TWB, SOC, and resilience by establishing a new model that integrates PERMA positive well-being theory [22], salutogenesis theoretical approach [74], resilience theory [24], and the impact of COVID-19 on higher education [13].

Regarding the first hypothesis, we examined whether COVID-19 stress is correlated with SOC, resilience, and TWB and found that such a correlation was negative and significant. This indicates that SOC, resilience, and TWB are negatively associated with COVID-19 stress; that is, teachers with a high level of SOC (manageability, comprehensibility, and meaning in life), resilience, and positive well-being experience reduced effects of COVID-19 stress. This is corroborated by the findings in the scientific literature [45,53–57,62,63,90]. Overall, SOC, resilience, and positive well-being as integrated resources are useful in minimizing COVID-19 stress in one's professional and everyday life.

For the second hypothesis, we examined the levels of the studied variables using standardized cut-off points and average means and standard deviations. We found that the highest scores were for resilience (26.7 out of 30, SD = 3.81) and SOC (56.2 out of 91, SD = 8.60), followed by TWB (72.5 out of 150, SD = 20.83), while the lower score was for COVID-19 stress (mean = 22.86, SD = 10.14). These findings support Hypothesis 2 and are consistent with those of [53,56], argued that SOC is a psychological resource that views existing conditions as manageable, comprehensible, and meaningful and has been highly associated with greater stress resistance and better psychological health. The results also showed that people with more positive emotions than negative ones experience a flourishing, joyful life; feel a sense of fulfillment; and effectively recover from stressful situations as well as lead better lives [62,63,91].

In addition to SOC, resilience as a positive psychological resource [62,67,92] and the PERMA positive well-being model [31,78] have potential roles in reducing stress and depression [45]. According to [45], higher levels of resilience and optimism are strongly associated with higher levels of positive well-being or happiness and lower stress levels. Meanwhile, studies on the relationship between resilience and teachers' positive well-being during the pandemic are crucial. According to [13], the COVID-19 pandemic has affected teachers' work-life balance and well-being. Therefore, this study proposes building the resilience of higher-education teachers as a core strategy to reduce their stress or frustration due to COVID-19 and boost their positive well-being. Other scholars have observed a high interaction between stress and resilience and well-being, and that resilience and low stress were influential predictors of well-being [65].

The third hypothesis tested whether COVID-19 stress negatively affects SOC, resilience, and TWB. This study found that COVID-19 stress is a negative predictor of SOC, resilience, and TWB, supporting Hypothesis 3. These findings are consistent with those of other scientific studies [25,27,45]. These results indicate that university teachers who exhibit high levels of SOC, resilience, and positive well-being experience lower COVID-19 stress.

The fourth hypothesis examined whether SOC and resilience are positive and significant predictors of TWB. According to the results, SOC is a significant positive predictor of TWB, which is supported by the literature [53,56,57]. Antonovsky's salutogenic theory also showed that generalized resistance and the use of personal resources help decrease stress and depression levels [35,57].

In addition, this study found that resilience is a significant positive predictor of TWB. Consistent with our findings, several works also mentioned the protective role of resilience. For example, resilience leads to several benefits such as physical or psychological integration [24,45,47–49,52,53], lower levels of COVID-19 stress [47], having positive emotions rather than negative ones [50], psychological adjustment [45], and developing hope and meaning in one's life [38]. Higher resilience is also associated with higher levels of positive happiness, lower levels of stress, resistance to disease, and recovery from adversity [45,51], and reduces the negative impacts of stress and enhances an individual's well-being [65].

The fifth hypothesis assessed COVID-19 stress as a predictor of TWB through SOC and resilience (see Figure 2). We found that SOC and resilience fully and significantly mediated the relation between COVID-19 stress and the TWB model, supporting Hypothesis 5. Other scientific findings were also consistent with those of this study [25,27,45,53,54,65]. Specifically, [45,56,57] found that resilience and SOC are the best personal resources that can safeguard ego strength and lower stress levels and depression.

The partial mediation model also confirmed whether SOC mediates the relation between COVID-19 stress and TWB (see Figure 3). This implies that higher SOC leads to lower COVID-19 stress and better TWB. The existing literature supports our findings and indicates that SOC protects and mediates adverse life experiences and positive well-being [53]. In fact, [56] also found that SOC had a buffering effect on public psychological health during the COVID-19 pandemic. Antonovsky's salutogenic theory also showed that generalized resistance and the use of personal resources help reduce stress and depression levels [57,90,92].

Finally, this study determined whether resilience plays a mediating role in the relationship between COVID-19 stress and TWB. We found that COVID-19 stress had an indirect, negative, and significant effect on TWB through resilience (Figure 4). Consistent with our findings, and as stated previously, several studies have discussed the protective role of resilience [45,49–51,61,62,92].

## 5. Conclusions

This study examined the direct impact of COVID-19 stress, SOC, and resilience on TWB as well as the mediation role of SOC and resilience between COVID-19 stress and TWB using the bootstrapping approach in SEM. We also determined the construct validity and reliability, as well as the measurement equivalence of the PSS-10-C, SOC-13, BRS, and PERMA Profiler Questionnaire, using CFA.

Even though the instruments were cross-culturally validated, construct validity using CFA, discriminant validity, convergent validity, composite reliability, and measurement invariance were performed and confirmed in this study.

As a result of the current findings, resilience and SOC had the highest scores, followed by TWB and COVID-19 stress. This is supported by the literature, which shows that higher SOC and resilience scores, as well as the PERMA positive psychology model, resulted in better stress coping and the maintenance of an individual's well-being. The findings of this study confirmed that SOC and resilience positively predict TWB and act as mediators between COVID-19 stress and TWB. These findings suggest that SOC, resiliency, and the PERMA positive well-being theory could all help to lower COVID-19 stress levels.

Thus, positive psychology intervention and prevention approaches that use resilience and SOC as positive resources to help teachers flourish in life and develop high resilience and SOC to nurture their well-being should be designed. Furthermore, this study demonstrated that higher levels of resilience, SOC, positive well-being among teachers according to PERMA, and lower levels of COVID-19 stress constitute a novel integrated model, are critical to overcoming existing problems, and are the best predictors for other professions such as health. Well-being is broad, and each professional task is different; therefore, we suggest that research be conducted in different contexts using this model to address employee well-being issues. Hence, this model will be applicable to clinical and other organizations.

Overall, this study provides practitioners and researchers who wish to work in such fields with mediation models that are based on the most recent academic research as well as manageable, time-saving, and more accurate psychometric tools, thereby bolstering efforts to comprehend the COVID-19 pandemic's effects and develop effective protective measures and interventions to increase TWB.

**Supplementary Materials:** The following supporting information can be downloaded at: https://www.mdpi.com/article/10.3390/ejihpe13010001/s1, Supplementary File: Both English Version and Amharic Translation instruments used for this study.

**Author Contributions:** Conceptualization, G.T.Z., B.T. and S.D.B.; data collection, B.T. and S.D.B., methodology, G.T.Z. and M.H., statistical analysis, G.T.Z.; material support, M.H. and F.S., writing—original draft preparation, B.T., S.D.B. and G.T.Z.; writing—review and editing, G.T.Z., M.H. and F.S. All authors have read and agreed to the published version of the manuscript.

**Funding:** This research received no external funding.

**Institutional Review Board Statement:** This study was granted an ethical approval letter (Ref. No. 26) from the institutional review board of the Doctoral School of Education, University of Szeged.

**Informed Consent Statement:** This study was performed according to the American Psychological Association (APA) and National Association of Psychology ethical standards for the treatment of human subjects. Data collection was anonymous and involved no identifying information and no medical treatment. Participants were informed that their participation was voluntary, that they could leave the study at any time, and that their data would be treated anonymously. In addition, they were informed that by starting the survey they would be considered to have read and accepted the informed consent.

**Data Availability Statement:** The data sets generated and analyzed during the current study are available from the corresponding authors, who are willing to share them upon request.

**Conflicts of Interest:** The authors declare no potential conflict of interest concerning this original research, authorship, and publication of this article.

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
