# Peer review of "COVID-19 Stress and Teachers Well-Being: The Mediating Role of Sense of Coherence and Resilience"

_ejihpe, doi:10.3390/ejihpe13010001_

Round 1

Reviewer 1 Report

I want to thank the authors and the Editorial Board for the opportunity to review the article submitted to the European Journal of Investigation in Health, Psychology and Education. The authors’ manuscript refers to a very important topic: teachers’ well-being during the COVID-19 pandemic. I believe it deserves its publication after some major changes.

Hypotheses: The current hypotheses are redundant. For example, H6 and H7 are represented in H5. Splitting H5 into two separate hypotheses is not necessary, the only difference between them is pointing out that the mediation would be partial and not full. I believe that they should be rewritten.

Figures: In SEM, latent variables should be represented by circles and observed variables should be represented by squares/rectangles. See Hair et al. (2019) for more information. I suggest that authors change their figures to be in line with industry recommendations.

Table 1: I highly recommend that authors rework Table 1. Usually, the names of the variables are put in the first column and model fit indices for the particular scale are presented in one row. It would greatly increase this table’s readability.

Table 1: there is a mistake in the rule of thumb section for SRMR and RMSEA coefficients – the sign should be the opposite one.

Common Method Bias: Harman’s single-factor method is highly criticised since it is not the best at detecting CMB in variance-based SEM models (see Hair et al., 2019; Knock & Lynn, 2012). I highly recommend that authors calculate VIF coefficients values for the tested model in order to verify if CMB is absent.

Results: Authors did not test very important coefficients which are required in the analysis of latent models: construct composite reliability (CR; the extent to which the set of measured variables actually represents the theoretical latent construct that these variables are intended to measure), convergent validity and discriminant valid (measured by AVE coefficients; the extent to which the indices of a particular construct converge or share a high percentage of variance and the extent to which a construct really differs from other constructs respectively). Without them, based only on CFA results, it is impossible to verify if the suggested latent variables are reliable.

Results: I recommend that authors perform the invariance analysis for the verified model. It is highly recommended to detect if the hypothesised structure is the same among all tested groups. The measurement equivalence/invariance can be tested on three different levels

·         Construct invariance – verifies if the construct is the same in the compared groups

·         Metric invariance – tests if factor loadings are similar (allows to perform correlation analyses)

·         Scalar invariance – verifies if factor loadings and regression constants are similar (allows to test group differences)

Author Response

Manuscript ID: ejihpe-1942556

Article Type: Research Paper

Title: COVID-19 Stress and Teachers’ Well-Being: The Mediating Role of Sense
of Coherence and Resilience

CORRECTION REPORT

No.

Reviewer Code

             Review

Corrections made by the author (s)

1

Reviewer 1

Hypotheses: The current hypotheses are redundant. For example, H6 and H7 are represented in H5. Splitting H5 into two separate hypotheses is not necessary, the only difference between them is pointing out that the mediation would be partial and not full. I believe that they should be rewritten.

Dear reviewer,

Thank you very much for your professional reviews and exciting advice that make our article much better. We appreciate it.

Based on your suggestions, and this version of the study reduced hypotheses 6 and 7.

Besides, we included the hypothesis in the introduction section

2

Reviewer 1

Figures: In SEM, latent variables should be represented by circles and observed variables should be represented by squares/rectangles. See Hair et al. (2019) for more information. I suggest that authors change their figures to be in line with industry recommendations.

Thank you for your impressive and exciting comment. Based on your suggestions, we checked and revised the figures based on  Hair et al. (2019) recommendation. Therefore, all four Figures are corrected.

3

Reviewer 1

Table 1: I highly recommend that authors rework Table 1. Usually, the names of the variables are put in the first column and model fit indices for the particular scale are presented in one row. It would greatly increase this table’s readability.

Thank you for your interesting and valuable suggestions. We have reorganized it based on your recommendation (please see Table 4).

4

Reviewer 1

Table 1: there is a mistake in the rule of thumb section for SRMR and RMSEA coefficients – the sign should be the opposite one.

Thank you for your kind and helpful suggestions. In this revised manuscript, We are correcting it.

5

Reviewer 1

Common Method Bias: Harman’s single-factor method is highly criticized since it is not the best at detecting CMB in variance-based SEM models (see Hair et al., 2019; Knock & Lynn, 2012). I highly recommend that authors calculate VIF coefficients values for the tested model in order to verify if CMB is absent.

Thank you very much for your suggestion and critical comment. Based on your recommendation, we have added Tolerance and VIF to detect multi-collinearity (see Table 1). However,  there is a possible way to deal with common method biases. According to Philip M. Podsakoff et al. (2012) and Podsakoff et al. (2003), common method biases are essential and considered the following key points:(a) the content or face validity of each item evaluated by experts in the field before administering the instrument; (b) Informed consent was obtained from all participants and their identity coded anonymously; (c) some items were reversely scored; (d) the predictor and the criterion variables were taken from different sources and cultural contexts; (e) for the issue of measurement error, the factor variance was computed. These are included and considered in this study. Therefore, we believe that the presence of common method biases is an asset (strength) for the study and better to keep it. 

6

Reviewer 1

Results: Authors did not test very important coefficients which are required in the analysis of latent models: construct composite reliability (CR; the extent to which the set of measured variables actually represents the theoretical latent construct that these variables are intended to measure), convergent validity and discriminant valid (measured by AVE coefficients; the extent to which the indices of a particular construct converge or share a high percentage of variance and the extent to which a construct really differs from other constructs respectively). Without them, based only on CFA results, it is impossible to verify if the suggested latent variables are reliable

We appreciate and thank you for your insightful comment.

The revised version of this study  conducted CR, convergent, and discriminant validity (Using AVE and MSV) to ensure validity and reliability indices.

7

Reviewer 1

Results: I recommend that authors perform the invariance analysis for the verified model. It is highly recommended to detect if the hypothesized structure is the same among all tested groups. The measurement equivalence/invariance can be tested on three different levels.

Dear reviewer,

We thank you lot for your exciting recommendation.

We performed the four strategies of measurement invariance (the Configural or construct, the metric, the scalar and residual across the gender, experience and university types). See Table 5 and the analysis highlighted in green colour.

Note:

  • All the newly modified parts in the manuscript’s text are highlighted in green colour.
  • A Professional proofreading service did the proofreading.
  • The revised version was strictly following APA 7th Edition.

Reviewer 2 Report

Comments:

The study focused on the positive impacts of SOC and personal-level resilience on well-being of the teachers under the COVID-19 pandemic. It can provide useful insights for managements of educational organizations. The following comments need to be considered before the publication to improve the quality of the paper.

1.       It can be assumed that respondents’ socioeconomic attributes such as age, sex, and income can influence the results. Ideally, the model will be revised to consider those variables. If it is not feasible, authors need to mention why they did not consider them and what the possible results are if the socioeconomic attributes are considered in the analysis.   

2.       The originality of this research is still not clear, because positive influence of SOC and resilience on well-being is self-evident.

3.       The hypotheses need to be restructured. For example, Hypotheses 1 to 4 can be categorized into one hypothesis. Hypotheses 5 to 7 can be also categorized into one hypothesis or integrated with Hypotheses 1 to 4.

4.       The term, resilience, has broad meanings. Authors need to explain the specific meaning of “resilience” in the paper concisely in the beginning of abstract and main tex.

5.       The introduction section is redundant and the structure needs to be revised. For example, the term resilience is explained in different subsections. Subsections 1.3 and 1.4 can be integrated with 1.2. In the subsection 1.2, authors need to explain and highlight the difference and relationships between SOC and resilience. Because resilience is a broader concept than SOC, I think SOC can be included in resilience, if the resilience is defined properly. At least, it is obvious that SOC and resilience are not independent.     

6.       When SOC first appears in abstract and main text, “sense of coherence (SOC)” should be added in respective texts. As for TWB, authors need to add “teachers’ well-being”, when it first appears. Authors need to carefully check the abbreviations used in the paper. CFA and EFA are also mentioned without proper explanations.

7.       In the Line 99, it should be COVID-19 (coronavirus disease 2019), not coronavirus pandemic, and it needs to be mentioned earlier in the beginning of the introduction.

8.       In the method section, some results are described but it is not proper. All the results should be mentioned in the result section. The tables and figures placed in the current manuscript show some main results.

9.       The results might be influenced by the geographical locations and standards or levels of universities. At least, the number of universities surveyed can be added in 2.1.

10.    The method to distribute and collect the questionnaire sheets should be mentioned in the method section because it is important information to validate the process of the survey.

11.    English proofreading by editors who know scientific writing may be needed. In Line 222, “For instance” is not needed. A list of abbreviations should be added.

Author Response

Manuscript ID: ejihpe-1942556

Article Type: Research Paper

Title: COVID-19 Stress and Teachers’ Well-Being: The Mediating Role of Sense
of Coherence and Resilience

CORRECTION REPORT

Reviewer 2

1

Reviewer 2

It can be assumed that respondents' socioeconomic attributes such as age, sex, and income can influence the results. Ideally, the model will be revised to consider those variables. If it is not feasible, authors need to mention why they did not consider them and what the possible results are if the socioeconomic attributes are considered in the analysis.   

Dear reviewer,

Thank you for your professional and insightful reviews. We accepted your valuable comment and to prove the constructs' measurement model. This version of the study tested gender, experience and university type using four-stage measurement invariance. In addition, we put it in the limitation of the study section for future study

2

Reviewer 2

The originality of this research is still not clear, because positive influence of SOC and resilience on well-being is self-evident.

Thank you for your comment. However, as far as the authors' knowledge, no study connects the SOC, resilience and Positive PERMA well-being model. In addition, there is no meditation study linking COVID-19 and Positive well-being. Therefore we believe that SOC and resilience played this mediation role. The originality and novelty of the study are clearly stated in the introduction section.

3

Reviewer 2

.       The hypotheses need to be restructured. For example, Hypotheses 1 to 4 can be categorized into one hypothesis. Hypotheses 5 to 7 can be also categorized into one hypothesis or integrated with Hypotheses 1 to 4.

Thank you for finding out this. The study hypotheses are restricted, and hypotheses 6 and 7 merged with hypothesis 5. However, the level or status, correlation, direct effect and indirect effect are treated separately

Thank you very much!

4

Reviewer 2

The term, resilience, has broad meanings. Authors need to explain the specific meaning of "resilience" in the paper concisely in the beginning of abstract and main text.

Dear reviewer,

Thank you very much for your suggestion. Here the term resilience depends on Masten and Reed (2002) and Davison, 2003. theoretically, resilience and a sense of coherence are distinct constructs.

5

Reviewer 2

The introduction section is redundant and the structure needs to be revised. For example, the term resilience is explained in different subsections. Subsections 1.3 and 1.4 can be integrated with 1.2. In the subsection 1.2, authors need to explain and highlight the difference and relationships between SOC and resilience. Because resilience is a broader concept than SOC, I think SOC can be included in resilience, if the resilience is defined properly. At least, it is obvious that SOC and resilience are not independent.     

Thank you very much for your suggestion. Here the term we separated 1.2, 1.3 and 1.4 because of their function and relationship. Readers easily identify the relationship between SOC and well-being, resilience and well-being etc. our study nature is associational, and each variable must be connected. However, SOC and resilience are distinct even though their correlattion is not high (see Table 2). In addition, we have checked the merged measurement model to check such problems.

6

Reviewer 2

When SOC first appears in abstract and main text, "sense of coherence (SOC)" should be added in respective texts. As for TWB, authors need to add "teachers’ well-being”, when it first appears. Authors need to carefully check the abbreviations used in the paper. CFA and EFA are also mentioned without proper explanations.

Dear reviewer,

Thank you very much for your critical comments.

We have checked and revised all.

7

Reviewer 2

In the Line 99, it should be COVID-19 (coronavirus disease 2019), not coronavirus pandemic, and it needs to be mentioned earlier in the beginning of the introduction.

Dear reviewer,

Thank you very much. We have now corrected it from the COVID-19 pandemic to coronavirus diseases. We appreciate and happily received your comments and suggestion

8

Reviewer 2

In the method section, some results are described but it is not proper. All the results should be mentioned in the result section. The tables and figures placed in the current manuscript show some main results.

Dear reviewer,

We thank you lot for your interesting comments.

Currently, we have rechecked and interpreted, plus added some valuable results. Kindly refer it the main text highlighted in the green colour

9

Reviewer 2

The results might be influenced by the geographical locations and standards or levels of universities. At least, the number of universities surveyed can be added in 2.1.

Dear reviewer,

Thank you very much. We have now clearly mentioned the sample universities and locations clearly.

10

Reviewer 2

The method to distribute and collect the questionnaire sheets should be mentioned in the method section because it is important information to validate the process of the survey.

Dear reviewer,

We are appreciated your valuable comment. We included and justified how to collect samples and from which universities, etc, in the method section.

11

Reviewer 2

English proofreading by editors who know scientific writing may be needed. In Line 222, “For instance” is not needed. A list of abbreviations should be added.

Dear reviewer,

Thank you very much for your suggestion.

Based on your suggestion and to improve the language issues of the manuscript, we did proofreading by the Proof-Reading-Service of ENAGO and attached the certificate below. I hope so; the grammatical, punctuation and spelling errors might be solved.

Thank you very much for your detailed comments.

Note:

  • All the newly modified parts in the manuscript’s text are highlighted in green colour.
  • A Professional proofreading service did the proofreading.
  • The revised version was strictly following APA 7th Edition.

Round 2

Reviewer 1 Report

I'm satisfied with the changes made by the authors.

Author Response

Manuscript ID: ejihpe-1942556

Article Type: Research Paper

Title: COVID-19 Stress and Teachers’ Well-Being: The Mediating Role of Sense
of Coherence and Resilience

Dear reviewer,

We appreciate your time, effort, and professional expertise. Your professional and insightful comments provided us with an enormous opportunity to revise our manuscript. We are grateful for your kind words.

Note:

  • The ENAGO proofreading service organization and Proofreading.com did a Professional proofreading service.

Reviewer 2 Report

I think the authors replied to my comments and the paper has been improved.   However, the following points need to be improved or revised before the publication, I think;   >Adding notes for the figures to explain e1, CO1, etc. >Discussion needs to be revised based on the revised results including new findings about demographic indicators. The number of hypotheses is not seven anymore.  >Some numbers in the figures can not be seen clearly.

Author Response

Manuscript ID: ejihpe-1942556

Article Type: Research Paper

Title: COVID-19 Stress and Teachers’ Well-Being: The Mediating Role of Sense
of Coherence and Resilience

CORRECTION REPORT

Response to the reviewer's comments

Reviewer #2
I think the authors replied to my comments, and the paper has been improved.   However, the following points need to be improved or revised before publication.
Comment 1. Adding notes for the figures to explain e1, CO1, etc.

Response: Dear respected reviewer, Thank you for your valuable suggestions once more. Your suggestions improved our research paper significantly. In response to your comment, we have added detailed notes to explain the letters that represent latent variables in the Figure, such as CO4, P1, and so on. All changes are recorded in the track change section of the manuscript.

Comment 2. Discussion needs to be revised based on the results, including new findings about demographic indicators.

Response:   Respected reviewer, thank you once more for your insightful comments. In response to your suggestion, we have discussed the newly added demographic indicators in this manuscript. Of course, the primary goal of checking demographic indicators is to ensure measurement equivalence across groups and the instruments' psychometric properties. The description is discussed on page 7, and the conclusion is on page 10.

Comment 3. Some numbers in the figures can not be seen clearly.

Response: Dear reviewer, we have revised the visibility of the numbers in the figures based on your valuable feedback. Figures 2, 3, and 4 show the necessary changes.

Note:

  • The ENAGO proofreading service organization did a Professional proofreading service.
